# Low-Cost Devices for Three-Dimensional Cell Aggregation, Real-Time Monitoring Microscopy, Microfluidic Immunostaining, and Deconvolution Analysis

**DOI:** 10.3390/bioengineering9020060

**Published:** 2022-02-03

**Authors:** Andreas Struber, Georg Auer, Martin Fischlechner, Cody Wickstrom, Lisa Reiter, Eric Lutsch, Birgit Simon-Nobbe, Sabrina Marozin, Günter Lepperdinger

**Affiliations:** 1Department of Biosciences and Medical Biology, University Salzburg, A-5020 Salzburg, Austria; Andreas.Struber@plus.ac.at (A.S.); Georg.Auer@plus.ac.at (G.A.); MFischlechner@gmail.com (M.F.); Cody.Wickstrom@stud.sbg.ac.at (C.W.); Lisa.Reiter@sbg.ac.at (L.R.); Eric.Lutsch@plus.ac.at (E.L.); Birgit.Simon@plus.ac.at (B.S.-N.); Sabrina.Marozin@plus.ac.at (S.M.); 2Sarcura GmbH, A-3400 Klosterneuburg, Austria

**Keywords:** 3D-cell biology, teflon, hanging-drop culture, delta-kinematic, rapid-prototyping, histology, whole-mount staining

## Abstract

The wide use of 3D-organotypic cell models is imperative for advancing our understanding of basic cell biological mechanisms. For this purpose, easy-to-use enabling technology is required, which should optimally link standardized assessment methods to those used for the formation, cultivation, and evaluation of cell aggregates or primordial tissue. We thus conceived, manufactured, and tested devices which provide the means for cell aggregation and online monitoring within a hanging drop. We then established a workflow for spheroid manipulation and immune phenotyping. This described workflow conserves media and reagent, facilitates the uninterrupted tracking of spheroid formation under various conditions, and enables 3D-marker analysis by means of 3D epifluorescence deconvolution microscopy. We provide a full description of the low-cost manufacturing process for the fluidic devices and microscopic assessment tools, and the detailed blueprints and building instructions are disclosed. Conclusively, the presented compilation of methods and techniques promotes a quick and barrier-free entry into 3D cell biology.

## 1. Introduction

Experimental analyses in cell biology commonly build on 2D-cell cultivation and the corresponding analytical methodology [1,2]. Yet, in 1944, 3D-cell culture was already being employed to study gastrulation [3], thus forging a new paradigm that 3D models would lead to an even better understanding of the basic mechanisms of biology [4]. Hence, 3D analyses have been increasingly applied. In particular, the development of 3D-supporting technology is currently making considerable progress [5]. In this context, affordable technology, combined with the establishment of common standards for obtaining meaningful experimental results, is desired [6].

Cell aggregates are regarded differently from cells cultivated on synthetic substrates as they build up microtissues using interstitium that guides and influences the gradients of the nutrients and cellular waste [7]. Cells at the surfaces of 3D-organotypic cultures may show high proliferation rates. At the same time, cells reside in quiescence in a basal layer, while within the core structures, others may undergo programmed necrosis. Viewing and analysing such processes side-by-side allows for the addressing of open questions relevant to both the basic sciences and pharmacology [8].

In terms of analytical approaches, 3D-models built from human cells are anticipated to provide a better means for completing potency assays in clinical research [9]. Hence, enabling methodology is currently being developed, and various techniques for 3D culturing are being adopted, including microfluidics [10], in vitro fertilization and embryo manipulation [11], bio-printing with cells encapsulated into extracellular matrix hydrogels [12], and hanging drop tissue culturing [13,14].

The manipulation of a delicate, self-organized cellular object is challenging, laborious, and often difficult to accomplish. We here describe a platform for controlled cell aggregation within hanging drops. This can be combined with a delta-kinematic viewing stage, which can be equipped with video microscopes for the online monitoring of morphological aggregate characterization. In addition to that, we also included protocols for visualizing the expression patterns of markers, mounting the specimen, and spatial analysis in 3D. Our goal was to create technology that could be inexpensively and rapidly manufactured, as well as easily adapted to individual needs. Based on this novel invention, innovative 3D cell biological models can be rapidly and easily produced.

## 2. Materials and Methods

### 2.1. Device Construction and Manufacture

The following cell-analytical devices were constructed: (1) the “Teflon Hanging Drop Device” (THDD) together with a chip holder (Appendix A), (2) the “Humidifying Incubation Chamber” (HIC), (3 a, b and c) the “Twin Microwells” (TµW-PDMS and TµW-PMMA) (Appendix A) together with an auxiliary Devices for in chip fluid exchange (Appendix A ), and (4) the Delta-XYZ-kinematic inverted microscopic video stage (δ-M) (Appendix A). The THDD was placed inside the HIC, which resides on top of the δ-M for online monitoring. After the completion of cell aggregation, the THDD was aligned to the inlets of the TµW device, in which the cell aggregates were further analysed. The THDD was made of 2 mm thick, 24 × 60 mm wide polytetrafluoroethylene (Teflon). Holes were milled through the material at 12.000 rpm with the aid of a milling device (Qbot, MiniMill, Austria) equipped with a 3 mm 60° deburring tool (Hans Treiber, Henstedt-Ulzburg, Germany). Molds for PDMS casting and device production are described in detail in the Appendix A, PMMA manufacturing in Appendix A. CAD/CAM files, as well as blueprints, building plans and software for δ-M, are disclosed and shared under https://github.com/spoc-lab/ (accessed on 15 November 2021).

### 2.2. Cell Aggregation and Analysis

The cell line hFOB 1.19 (American Type Culture Collection, CRL-11372) was cultivated in a 100 mm tissue culture dish at 34 °C with ambient air supplemented with 5% CO_2_ in Dulbecco’s Modified Eagle Medium/Nutrient Mixture F-12 (DMEM/F12, Sigma-Aldrich, Taufkirchen, Germany; D6421) containing 10% FBS (Corning, COR35-079), 1% L-glutamine (Sigma, G7513), 1% penicillin-streptomycin (Sigma-Aldrich, P4333), and 0.3 mg/mL Geneticin (G418; Sigma, G8168). Saos-2, an osteosarcoma cell line (American Type Tissue Culture, Manassas Virginia; ATCC^®^ HTB-85™) was cultivated in Dulbecco’s Modified Eagle Medium, high glucose (DMEM-F12; Sigma, D6421) containing 10% FBS (Corning, COR35-079), 1% L-glutamine (Sigma, G7513) and 1% penicillin-streptomycin (Sigma-Aldrich, P4333). The cells were used for aggregation before reaching confluence. The cells were dispersed by the addition of trypsin/EDTA and resuspended in the culture medium at a concentration of 250 cells/µL. Drops were formed using a 20 µL cell suspension hanging from the THDD surface and then incubated at 34 °C within the HIC, which provided a fully humidified atmosphere. Cells were allowed to aggregate for 48 h during which spheroid formation was monitored using the remotely controllable δ-M. Images were taken of each well at 30 min intervals. Thereafter, hanging drops were transferred into the TµW by adding 20 µL of the culture medium through the top hole of the THDD. A 3D-printed gauge was used to align the positions between the THDD and the inlets of the TµW. 

### 2.3. Whole-Mount Immunofluorescence Protocol

The Teflon hanging drop device (THDD) was aligned with the inlet of the twin micro-well staining chambers with the help of 3D printed holders (Appendix A). Next, the hanging drops with cell aggregates were directly dropped into a staining chip by adding another 20 µL of the culture media through the upper hole onto the hanging drop chip. Since the twin microwell device contains a glass floor, cell aggregates can be cultivated and examined using microscopy. The removal and addition of solutions was always accomplished with the use of suction through the microchannels using either a pipette tip or the suction device, leaving the cell aggregates undisturbed. 

Cell aggregates were fixed with 4% paraformaldehyde at room temperature for 30 min. After fixation, three washing steps using PBS were performed for 5 min to remove excess paraformaldehyde. Conventional immunohistological analysis of the fixed spheroids was performed as described in detail in Appendix A. For whole-mount staining, specimens were first permeabilized using 0.5% Triton X-100 solution in PBS for 15 min at room temperature. Thereafter, a blocking step was performed in 1% Bovine Serum Albumin (BSA), 0.2% Triton X-100, 0.05% Tween 20 and 10% Fetal Bovine Serum (FBS) in PBS at room temperature for 60 min to prevent non-specific antibody binding. Anti-Lamin A/C (Santa Cruz Biotechnology, sc-376248) or anti-KI67 (Santa Cruz Biotechnology, sc-23900) were diluted in a blocking solution to 2 µg/mL according to the manufacturer’s instructions. Cell aggregates were incubated overnight at 37 °C under gentle shaking. The following day, the cell aggregates were washed three times with PBS for 5 min each and subsequently incubated in the dark with the secondary antibody anti-mouse Alexa Fluor 555 (Cell Signaling, 4409S) at 37 °C for 4 h under gentle shaking. After three PBS rinses, the nuclei were counterstained with 10 µg/mL of Hoechst 33342 (Sigma, 14533) in PBS for 5–10 min at room temperature in the dark. Before the image acquisition, a final three-step PBS-rinse was performed and the spheroids were left in the PBS for microscopy.

### 2.4. Image Acquisition and Deconvolution Microscopy

Digital micrographs were acquired using the delta kinematic stage equipped with a video microscope (Bysameyee, https://www.bysameyee.com (accessed on 15 November 2021)) (for details see Appendix A), and with a widefield microscope (Leica DMi8 controlled by LAS X software 3.4.2). To acquire an image stack, endpoints in the z-axis of the specimen were first assessed; 40 µm was subtracted in either direction in order to obtain a slice in the central region of the spheroid. Step size was enhanced using LAS X software for optimal resolution according to the layer numbers of the stack. Image deconvolution was performed using Huygens Essential Software (Scientific Volume Imaging, Hilversum, The Netherlands).

## 3. Results

### 3.1. Formation, Handling, and Monitoring of Cell Aggregates in Hanging Drops 

Cells of mesenchymal origin may readily aggregate when incubated within a drop of culture medium hanging from a horizontal surface. For this special purpose, various types of materials were tested. Notably, only surfaces which exhibit relatively high hydrophobicity allow for stable hanging drop cultivation. We obtained the best results using Teflon, which could also be easily processed by milling. In order to support hanging drop formation, holes were drilled into a 2 mm Teflon plate with the aid of a 60° deburring tool such that a very small inlet hole for cell suspensions was generated on the top side of the plate, while a tapered borehole was generated from the bottom side (Figure 1A and Appendix A). When 20 µL was pipetted through the inlet of the THDD, a stable drop formed, which was then incubated for a period of up to one week (Figure 1B). Once the drop was established, a cell suspension containing 250 cells/µL was applied. Within two days, the cells built a stable cell aggregate, also called a spheroid (Figure 1C). In order to protect a hanging drop from drying out, a humidified incubation chamber (HIC), consisting of three parts, was manufactured using a 3D-printer. The upper and lower parts include a rectangular opening into which glass slides have been glued to allow observation of the cells during spheroid formation (Figure 1D,E). The thermal conductivity of the HIC walls was enhanced using thermoplastic filament printing material containing 20% bronze (for details, see Appendix A). The bottom section of the HIC was flooded with water, covering the entire floor, thus providing a humidified atmospheric condition. The Teflon plate holder was placed into the HIC and covered by the lid, its rim standing in the water. An assessment of the drop weight over an extended incubation period within the HIC revealed an evaporation rate below 2% per day.

In order to monitor cell aggregation during cultivation, an inverted microscopic stage was constructed based on delta kinematics (δ-M) (Figure 2; Appendix A). The mechanism is based on individual arms constructed using parallelograms that are arranged into a three-armed scaffold (Figure 2A). When the position of a parallelogram base is shifted by means of a stepper motor mounted to a threaded rod, the orientation of the microscope platform is maintained as movement is restricted to translation in the X, Y, or Z direction without rotation (comprising an XYZ stage). δ-M was further equipped with a commercially available, low-cost USB video microscope, which was completely computer-controlled via a web-based interface. The compact size, i.e., the small footprint and low weight of the video robot, allowed placement and computer-aided operation within the sterile environment of a fully conditioned cell culture incubator, which is unattainable using a common research microscope. Alternatively, building fully conditioned incubation chambers on a high-end microscope stage for research purpose could be rather costly. When placed into an incubator, the remote-controlled δ-M facilitated online monitoring of the HIC/THDD-assisted cell aggregation (Figure 2B). After forming a 20 µL hanging drop, the suspended cells sedimented in the course of 30 min (Figure 3A). The hFOB cells aggregated within 12 h (Figure 3B) and matured into delicate structures (Figure 3C) before compacting into rigid spheroids after 45 h (Figure 3D). The Saos-2 cell suspension prevented spheroid generation (Figure 3E,F).

### 3.2. In-Chip Fluidic Manipulation of Cell Aggregates

The handling and treatment of delicate cell aggregates is difficult, often resulting in the destruction and loss of the specimen, particularly when attempting to transfer a spheroid by means of pipetting. We therefore designed and manufactured a device that requires the superimposition of a hanging drop above a microwell (Appendix A). When a well (suitable for further incubation, staining, or microscopic analysis) is placed underneath the hanging drop and additional medium is added to the drop through the upper opening, the drop falls off. This transfers the spheroid into the microwell in a contactless fashion, thus greatly reducing the risk of damage and loss.

Next, twin microwells (TµW) were designed and manufactured in two different variations (TµW-PDMS and TµW-PMMA). Due to the employed material and the manufacturing method, the design, size, and format of the TµW-PMMA can be rapidly adapted (Appendix A). TµW-PDMS can also be easily equipped with pumps and integrated into a fluidics environment. The chip floors were sealed with a high-performance light microscopy coverslip. TµW-PDMS was made of polydimethylsiloxane (PDMS) comprised of cross-sectional microchannel dimensions of 60 × 100 µm or 60 × 400 µm to connect the well structures (Appendix A). For the removal of buffers and staining solutions, a suction device was constructed (Appendix A). When connected to a water jet pump, solutions can be withdrawn simultaneously from many wells. The cell aggregates are normally much larger than the channel dimensions; hence, buffer exchange can be easily accomplished by merely applying suction through the channels (Figure 4A). TµW-PMMA were made of acrylic glass (PMMA) that was laser-cut and bonded to a coverslip (Figure 4B, Appendix A). Most double-sided adhesive tapes exhibit strong autofluorescence; however, this is not the case with dry photoresist foils. We also found that this material exhibited strong enough adhesive properties under heat and pressure; thus smooth PMMA surfaces can be easily sealed using microscopic cover glasses. Here, the conceived microwells were actually connected by a very wide channel which can be locked and narrowed similar to a floodgate. The barriers were constructed such that the bottom side exhibited a 50 µm wide depression. Hence, by pinning a barrier piece between the primary structure, two microwells were demarcated while remaining connected by a narrow channel. TµW-PMMA fulfils the same function as TµW-PDMS, but is more versatile since the design and sizes of the barrier pins can be adjusted. We were able to show that cell aggregates can be successfully cultured within the TµW-PDMS. The process of media exchange and the addition of solutions was greatly eased, and compared to direct manipulation of spheroids in a single microwell, the procedure was conducted without the loss of or damage to the delicate structure (Figure 4C). Spheroids cultivated in the TµW-PMMA underwent rapid cell death (Figure 4D). The course was comparable to otherwise viable spheroids treated with 10 µM staurosporine. 

### 3.3. Whole-Mount Immunofluorescence Analysis of Cell Aggregates

Analyses of cell aggregates, spheroids, and organoids are often carried out by applying fluorescent dyes and antibodies (Figure 5). In order to carry out a successful staining in the TµWs, first, a reliable protocol for whole-mount immunofluorescence with hFOB cells was independently established. Based on that, protocols could be adjusted to reliably carry out staining. The staining results from whole-mounts are often debated. Large biomolecules, such as antibodies, might be hindered in reaching the deep layer. In turn, the background could be high due to insufficient washes. By following the herein presented protocol, we were able to show that carrying out whole-mount staining yields results comparable to histological analysis (Figure 5A,B). We compared immunological staining of paraffin-embedded spheroid sections and whole spheroids by applying fluorescently labelled anti-Ki-67, a widely used proliferation marker. The nuclei were counterstained with the nuclear dyes DAPI or Hoechst. Cells located in the outer layers of the cell aggregate express Ki-67 indicating proliferation. Staining performed on 3 µm histological sections revealed few Ki-67 positive cells in deeper layers. Whole-mount staining using anti-Ki-67 showed comparable results, thus strengthening the belief in a wide applicability of the analytical technique in combination with the herein described devices (Figure 5B). Images of whole mounts are blurred due to scattered light. Due to its higher sensitivity and dynamic range, it is generally accepted that widefield deconvolution microscopy is superior to other 3D-microscopic imaging techniques [15]. We therefore recorded a series of images of the whole mounts along the optical Z-axis with the aid of a widefield microscope and applied deconvolution image analysis. Working along this line, images could be restored yielding digital images, which provided the means for the quantitative analysis of expression patterns within biological 3D-objects which are not fully translucent (Figure 5C).

## 4. Discussion

Due to surface tension, hanging drop cultures can be easily suspended on a synthetic surface. Most often, a small volume of liquid is poured onto a dish, which is then inverted. Cells suspended in a mitogen-bearing culture medium proliferate while aggregating at the bottom of the drop [16,17,18,19]. Apparently, this type of hanging-drop culture exhibits limitations, as cultivation time, media change, or drug treatment is largely restricted. Manipulation, such as the passaging or staining of the delicate cell aggregates, is also a difficult undertaking, scarcely performable by robotic means. We also noted that the wetting of the cell culture plastic during cultivation slowly alters the drop geometry, thereby forming a very shallow pond-like hanging basin similar to results obtained using embryoid body cultures [20]. Previously, these problems have been addressed by using hydrophobic material such as PDMS [21], or by applying surface coatings, which in due course may compromise cell physiology [22]. 

We therefore took utmost care to use materials suitable for this purpose, Teflon in particular [23]. Using the proper materials, combined with the methods described, will support and thereby promote the production and implementation of low-cost micro-fluidic methods in most academic settings. The devices described here can be easily manufactured with commonly available tools and machines, e.g., in makerspaces at academic institutions, or in openly accessible engineering labs of private enterprises or communities. The materials are inexpensive and can be purchased without restrictions. Similar protocols have been communicated for specific areas of research and development in need of specifying standards of implementation for a broader use of this technology [24].

After several unsuccessful attempts using different sorts of synthetic materials such as PDMS or PMMA along with various surface coatings, hanging drop devices were made of polytetrafluoroethylene, commonly called Teflon. The completely inert and water-repellent material allows for easy machining using milling tools [23]. By cutting conical cavities, the stability of the drops was greatly enhanced. By drilling through a thin Teflon sheet, drops can be built and manipulated using the convenient pipetting of solutions and suspensions from above. The drop geometry also remained stable in the THDD when incubation took place in a fully humidified environment. Moreover, trans-illumination is possible in the THDD, hence live-cell-microscopy is greatly facilitated. Combining the THDD with online monitoring now enables the studying and assessing of the live kinetics and dynamics of characteristic changes of different cell types during aggregation and thereafter. Also, co-aggregation experiments, for instance using endothelial cells in order to monitor and survey vessel formation under changing conditions, become feasible. Notably, during cultivation, the addition of bioactive factors to the hanging drop through the upper THDD opening is possible in an undisturbed fashion. Ultimately, due to the simple design and manufacturing of the THDD, crowded side-by-side cultivation of drop cultures and the displacement of cell aggregates through automated handling is feasible [19]. Due to the inert nature of Teflon, the THDDs are re-useable, again reducing the cost and relieving budgets. 

Once aggregates are formed, subsequent cultivation and particular treatment procedures are often carried out in microwells exhibiting ultralow binding surface coatings. In order to ease in-line control, the interior of the well should be amenable to widefield inverted microscopy. The transfer, media change, and manipulation of aggregates often results in damage or loss. To ease the handling of spheroids, and to also offer means for potential automation approaches, the TµWs were designed with minimal risk of accidentally aspirating aggregates during cultivation and subsequent whole-mount phenotyping. We were able to show that cell aggregates can be successfully cultured within the TµW-PDMS with greatly reduced effort in regards to media exchange. Unfortunately, spheroid cultivation within the TµW-PMMA induced cell death rather quickly, comparable to specimens treated with staurosporine, a cell-permeable, ATP-competitive inhibitor of protein kinases, which leads to cell death. Despite this drawback, the benefits of the TµW-PMMA device are (i) low-cost production by means of widely available equipment, (ii) fast redesigning potential regarding the adjustment of the channel dimensions, (iii) expansion to virtually any size and microwell number, (iv) reliable fixation and staining abilities for the specimen and eventually, (v) sealing with a mounting medium and a coverslip.

For aggregate-testing in a well, we employed a cell line of mesenchymal origin, human fetal osteoblasts, (hFOB 1.19), which exhibit strong cell interactions to form a stable aggregate [24]. In contrast, the Saos-2 cells refrained from aggregation. Testing the devices with cells that normally firmly adhere to plastic indicated that the microwells can also be utilized to analyse other cell types since eventually, all surface structures presented little to no binding interfaces to the mesenchymal cell aggregates. When disregarding this issue, cell aggregates were captured in the corners and angled nooks of the wells or channel outlets. Attempting to remove stuck aggregates often led to damage or destruction. The TµW-PMMA is particularly versatile in this respect. Notably, the geometries of the TµWs have also been optimized to hold only small volumes, which is of considerable advantage when expensive reagents need to be used in medium-to-high through-put analyses. The small footprint of a single twin microwell structure allows for the production of analytical devices that hold many spheroids, thereby promoting the parallelization of 3D-cell aggregate analytics.

Eventually immune-phenotyped cell aggregates can be mounted for microscopic analysis within the TµWs. This needs to be emphasized, as we frustratingly experienced that immunohistological and microscopic analyses of cell aggregates, besides being time-consuming and expensive, most often result in either lost or irreversibly damaged specimens [25]. Apparently, the TµW can also be employed for high-performance laser scanning or spinning-disc microscopy. Yet, we were also keen on implementing methodology, which is affordable, but which also produces reliable 3D-cell biological analysis. Working towards this goal, we established a low-cost online monitoring platform. We also disclosed all blueprints and information on material processing and machining, as well as the software for controlling the final device in an automated fashion.

Lastly, we focussed on establishing gear, devices, and beacons that can be easily, and most importantly, inexpensively produced. The most elaborate and complex machinery used was the δ-M. Mechanical building blocks such as 3D-printed parts, threads, slide rails, screws magnets, and couplings cost around EUR 100, electronics and motors around EUR 200, and the video microscope was less than EUR 100. The materials and components necessary for building the devices in the context of this work were less than that, thus, even more cost saving. This approach may help in reducing the economical hurdles encountered when entering into basic 3D-cell biological research.

## 5. Conclusions

Many areas of experimental cell biology are currently shifting to implement beyond state-of-the-art 3D techniques in order to address open questions in basic research, drug screening, or regenerative medicine. Without the broad availability of affordable and accessible cultivation tools and assessment techniques, rapid progress is greatly impaired. Therefore, we present a collection of methods and devices that will allow every cell biologist to conduct 3D-experiments with little effort and great gain. The microfluidic cell devices and assessment tools can be rapidly manufactured at very low costs, and as disclosed here, easy adaptation to individual needs is both feasible and permissible. Using this instrumentation, side-by-side cell aggregates can be efficiently generated and analyzed in a simple, reproducible, and highly controllable fashion. Implementing this equipment also conserves both media and reagent, since the necessary volumes are in the microliter range. Furthermore, we were able to show that 3D-analyses of common markers can be visualized using microscopes commonly available in most cell biology laboratories. Preeminently, this procedure does not require laborious and time-consuming histological evaluation. By means of deconvolution microscopy, expression patterns of markers can be rapidly obtained and comparatively evaluated. Hence, the herein presented compilation of all the necessary equipment, methods, and techniques allows for a quick and easy entry into 3D-cell biology.

## Figures and Tables

**Figure 1 bioengineering-09-00060-f001:**
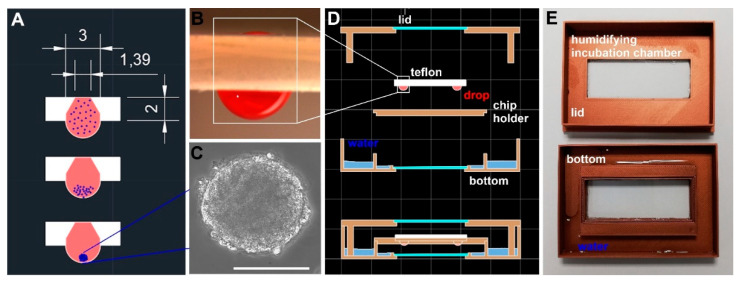
Devices supporting cell aggregation. (**A**): a schematic illustration/cross-section of the Teflon plate (white) utilized for the formation and incubation of hanging drops. Geometric measures of the drill hole and the plate thickness are given in mm. The process of cell aggregation (blue dots) within the cell culture media (red) is depicted as a sketch; (**B**) (side view): a picture of a hanging drop (for enhanced contrast, 20 µL of a deep red-colored solution was dispensed) held in place in the milled Teflon material; (**C**): a widefield microscopic image of stably aggregated osteoblasts forming a so called “spheroid” (the scale bar indicates 100 µm); (**D**): a schematic illustration of an incubation chamber including rectangular cutouts fitted with glass slides to allow for cell aggregation and observation. The chamber provides humidified atmospheric conditions, preventing the evaporation of the culture media during hanging drop formation at elevated cultivation temperatures; (**E**): the assembled incubation chamber.

**Figure 2 bioengineering-09-00060-f002:**
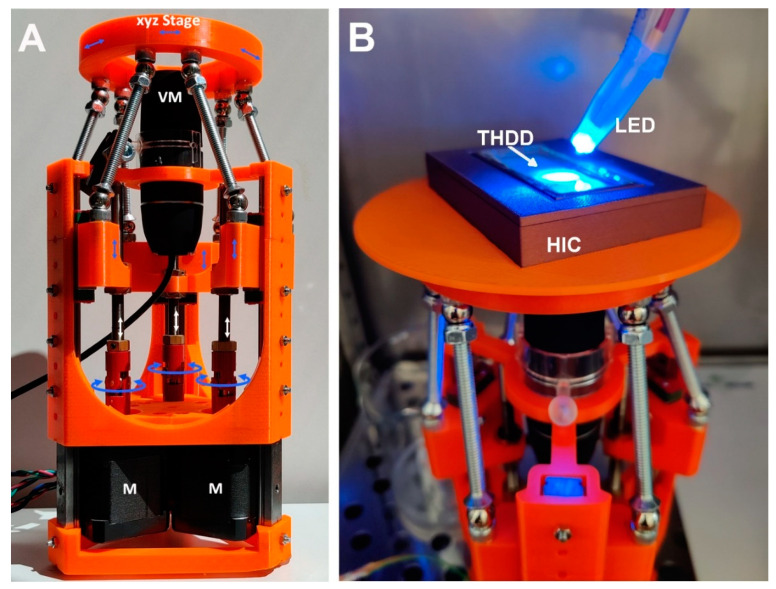
The delta kinematic microscopic stage/chip monitoring microscope, δ-M, with an incubation chamber. (**A**): For the on-chip observation of hanging drops, a microscope stage was 3D-printed and assembled, enabling controlled movements in a 3D fashion (XYZ stage), driven by accurate stepper motors (M); In the central position, a USB camera (VM) was mounted for acquiring photomicrographs from underneath (for a detailed description, see Appendix A); (**B**): a delta kinematic microscope stage holding a humidifying incubation chamber (HIC) containing a Teflon hanging drop device (THDD), completely illuminated by a blue-light emitting diode (LED).

**Figure 3 bioengineering-09-00060-f003:**
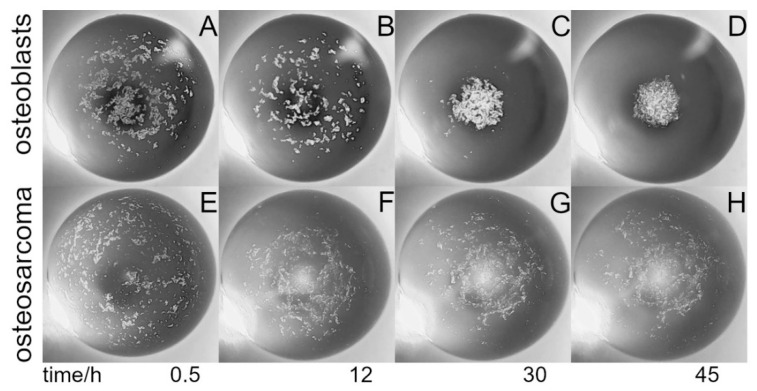
Hanging drop cell aggregation: Shown are time-series photos visualizing cell aggregation within a hanging drop contained 5000 cells in a 20 µL medium, viewed from underneath. The suspension was dispensed on the Teflon chip and incubated within the humidified chamber for indicted times. The images were acquired using the delta kinematic stage equipped with a commercially available low-cost video microscope (see Figure 2). The cell aggregation shown here was performed using human fetal osteoblasts, hFOB1.19 (**A**–**D**), and human osteosarcoma cells, Saos-2 (**E**–**H**). The field of view is 3 mm.

**Figure 4 bioengineering-09-00060-f004:**
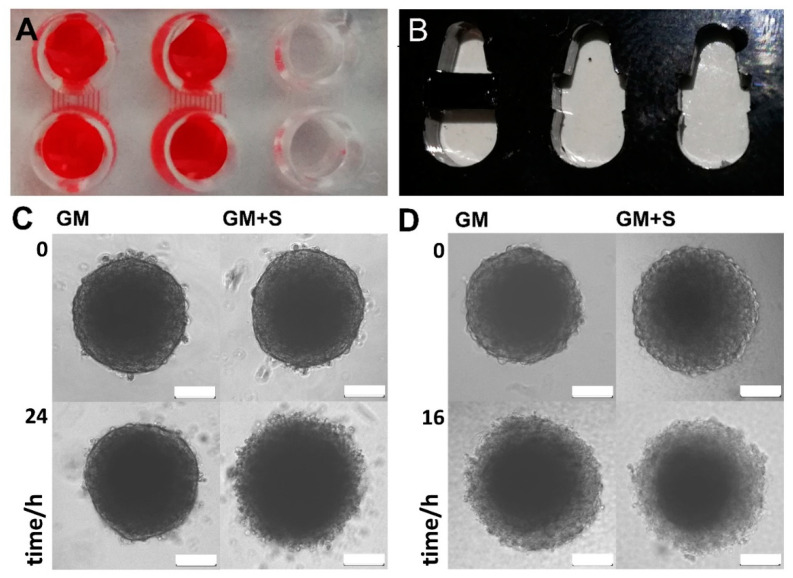
Twin-microwell 3D-cell aggregate analysis: (**A**): a photograph of a PDMS staining chip (top view) along with the connecting microchannels; in order to enhance the contrast and view the microfluidic channels, red solution was added to two chambers. The diameter of the wells is 3 mm; (**B**): a photograph of a PMMA staining chip (top view). The diameter of the larger (bottom) well is 3 mm; (**C**,**D**): microscopic images of spheroids cultured and treated in TµW-PDMS (**C**) and TµW-PMMA (**D**). Spheroids made from human osteoblasts, hFOB1.19 were cultivated in growth medium (GM), or in growth media containing 10 µM staurosporine (GM + S), a cell-permeable, ATP-competitive inhibitor of protein kinases that leads to the induction of cell death; the scale bars indicate 100 µm.

**Figure 5 bioengineering-09-00060-f005:**
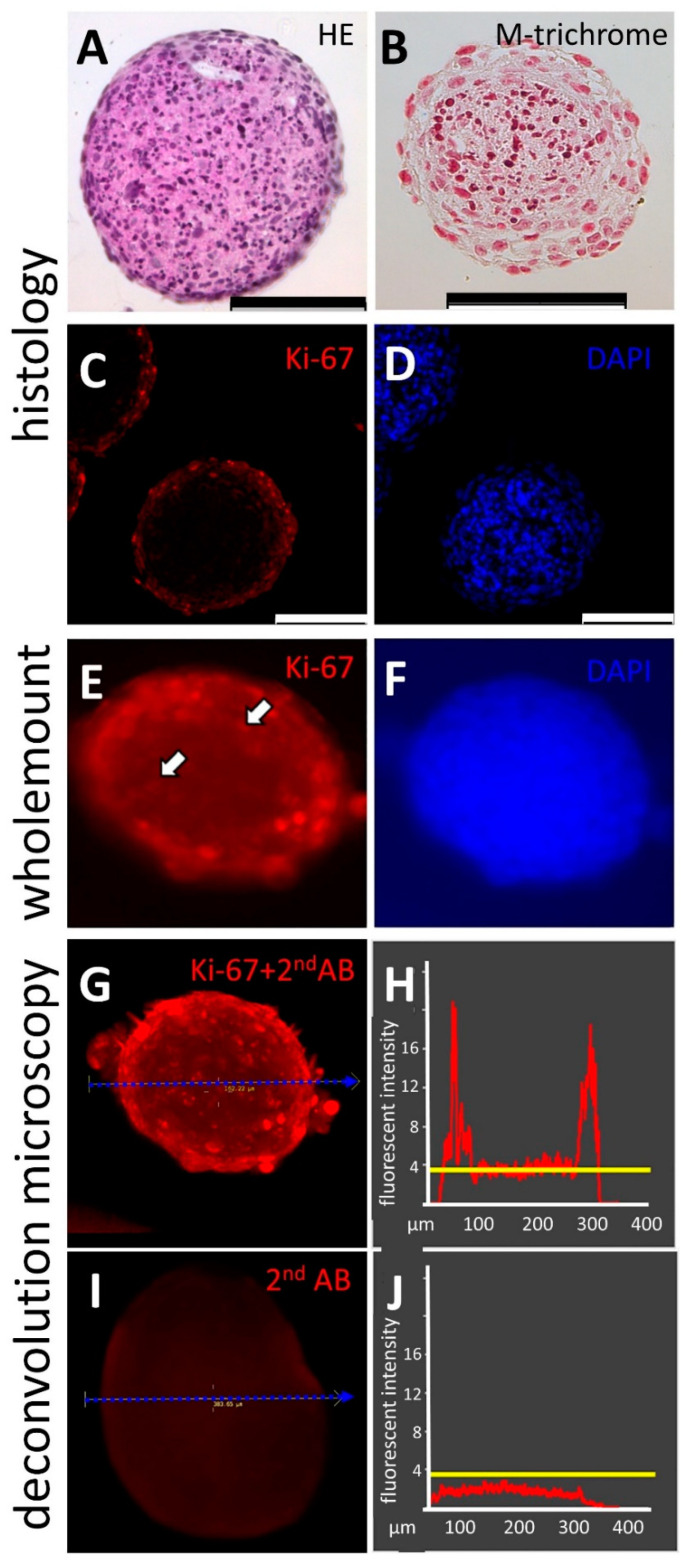
(**A**–**D**): the histological analysis of cell aggregates made from a human osteoblasts cell line, hFOB1.19 on 3 µm paraffin sections (Appendix A), (**A**): hematoxylin-eosin staining; (**B**): Masson’s Trichrome staining; (**C**): the immunohistological analysis of the proliferation marker Ki-67 (**D**): staining of cell nuclei with the DNA-binding fluorophore DAPI; (**E**,**F**): fluorescent imaging of a hFOB 3D whole-mount specimen at widefield optics before deconvolution, allowing direct comparison with the images derived from the 3µm histological sections shown in (**C**,**D**); the arrows in (**E**) point out Ki-67 positive nuclei present in the core region of spheroids; (**G**,**I**): deconvoluted digital images of hFOB cell aggregates stained as 3D whole mounts with anti-Ki-67 antibody and fluorophore-labeled secondary antibody (**G**) and, as a control, a spheroid stained with a solely labeled secondary antibody (2^nd^AB) (**I**). (**H**,**J**): Histograms depict pixel intensities along the dashed blue line as drawn in the corresponding images (**G**,**I**). The threshold level for histogram analysis was determined in the secondary AB-specimen (**I**). This set measure was applied for image analysis to discriminate signals from noise (strong yellow line in (**H**,**J**)). Shown here is spatial distribution and the corresponding analysis for the proliferation marker Ki-67 (**G**,**H**). The abscissae in H and J is the spatial distance in µm; the ordinate axis shows arbitrary fluorescence units; the scale bars indicate 100 µm.

## Data Availability

Further information on the design and manufacturing of the described devices can be found at https://github.com/spoc-lab/ (accessed on 15 November 2021).

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
