# Peer review of "Low-Cost Devices for Three-Dimensional Cell Aggregation, Real-Time Monitoring Microscopy, Microfluidic Immunostaining, and Deconvolution Analysis"

_bioengineering, 2022, doi:10.3390/bioengineering9020060_

Round 1

Reviewer 1 Report

Review comments:

Paper title: Low cost devices for three-dimensional cell aggregation, real-time monitoring microscopy, microfluidic immunostaining and deconvolution analysis

Authors: Andreas Struber, Georg Auer, Martin Fischlechner, Cody Wickstrom, Lisa Reiter, Eric Lutsch, Birgit Simon-Nobbe, Sabrina Marozin, Günter Lepperdinger

Summary:

The key premises of this work focused on an engineered approach which are cheap and adaptable for the hanging-drop spheroid culture applications. Overall the work is interesting but I feel the authors might have strayed a little off from their objectives of the paper which is a cheaper/ robust platform to enable spheroid culture and imaging with the most important part (the platform and the assembly) is placed at the supplementary. I think the author can rearrange some of the figures to make the work more impactful in delivering the message.

Detailed comments;

  1. Line 7 – 31: Too generic, consider shorten it and place the focus on the hanging drop technologies, standard approach and limitations so that you can build up toward the main claim at paragraph 3.
  2. Line 32-39: To be honest, the way the author wrote the key feature of the platform is a little tricky for me to catch on what it is the new platform can improve current limitations. I think authors can expand on the current limitations of hanging drop experiment (point 1) to link this better. I am interpreting it as a spheroid culture platform with in-built viewer that is cheap and easy to manufacture specifically to adapt to most biological studies. Consider rephrase the line 33-36 to flash this out?
  3. Methods section – image acquisition. Were all work imaged on the Leica microscope? If yes, why there is a USB camera the rig in the d-M platform? Any specific imaging parameters set for the assembled imaging platform?
  4. Results: SI figure 4 and the material build list should probably go into the main text in a new section. It is hard to tell what the platform is and how the parts come together. (Figure 1 does not tell much, and Figure 4 is confusing to follow)
  5. Line 143-146. I am not sure how does Figure 4 protects the loss of specimens during transfer? Consider expand more.
  6. Figure 2, label the parts for better clarity
  7. Figure 3, scale bar missing. Better to rotate the time axis 180 degrees for better reading. I am sure you can still measure the diameter of the spheroids in 48 hours for osteaoblast. Consider measuring them compare the aggregates from the osteosarcoma.
  8. Figure 4 – label for better clarity, missing and unclear scale bar.
  9. Figure 5 – which cell lines are these from? Scalebar blurry.
  10. Figure 5 H and J. These are the signal intensity between “stained” and ‘not stained” groups. From the aims in the introduction, authors should consider comparing signal between deconvolution vs wholemount vs histology group instead.
  11. Figrure 5 H and J, words and lines are blurry, axis label missing, use better resolution.
  12. Line 319-322 – I am not exactlu sure what if this paragraph about.
  13. 4 – label the parts
  14. For the aim of the paper, what is the current go-to method for viewing hanging drops? Why not just pipette/collect and view in microscopes?
  15. For the set out aims of the paper, might be good to compare the cost of manufacturing to the commercially available platforms too.
  16. I am not sure why is the immune-histological analysis is placed in supplementary 5.

Author Response

Comments from and to Reviewer 1:

Paper title: Low cost devices for three-dimensional cell aggregation, real-time monitoring microscopy, microfluidic immunostaining and deconvolution analysis

 Authors: Andreas Struber, Georg Auer, Martin Fischlechner, Cody Wickstrom, Lisa Reiter, Eric Lutsch, Birgit Simon-Nobbe, Sabrina Marozin, Günter Lepperdinger

COMMENT-1:  Summary:  The key premises of this work focused on an engineered approach which are cheap and adaptable for the hanging-drop spheroid culture applications. Overall the work is interesting but I feel the authors might have strayed a little off from their objectives of the paper which is a cheaper/ robust platform to enable spheroid culture and imaging with the most important part (the platform and the assembly) is placed at the supplementary. I think the author can rearrange some of the figures to make the work more impactful in delivering the message.

REPLY:  Thank you for this pointing out how to strengthen the message; I hope we were able to revise the manuscript in a way to make it more stringent according to the prime message.

Detailed comments;

COMMENT-2:Line 7 – 31: Too generic, consider shorten it and place the focus on the hanging drop technologies, standard approach and limitations so that you can build up toward the main claim at paragraph 3.

REPLY: thank you for pointing out. The first two introductory paragraphs have been rewriten now reading as follows (see revision - line 23-42):

Experimental analyses in cell biology build commonly on 2D-cell cultivation and corresponding analytical methodology [1, 2]. Yet, already in 1944 3D-cell culture was employed to study gastrulation [3], thus forging a new paradigm that 3D models will lead to an even better understanding of basic mechanisms in biology [4]. Hence, analyses in 3D have been increasingly applied. In particular, 3D-supporting technology is currently making considerable progress [5]. In this copntext, affordable technology together with common standards for gaining meaningful experimental results are desired [6].

Cell aggregates are regarded different from cells cultivated on synthetic substrates as they build up micro-tissues with interstitium that guides and influences gradients of nutrients and cellular waste [7]. Cells at the surfaces of 3D-organotypic cultures may show high proliferation rates. At the same time, cells reside in quiescence in a basal layer, while within core structures others may undergo programmed necrosis. Viewing and analysing such processes side-by-side allows addressing open questions in basic sciences, yet also of pharmacologic relevance [8].

In terms of analytical approaches, 3D-models built from human cells are anticipated to provide better means for potency assays in clinical research [9]. Hence as of now, enabling-methodology is being developed, and various techniques for 3D culture are being adopted including microfluidics [10], in vitro fertilization and embryo manipulation [11], bioprinting with cells encapsulated into extracellular matrix hydrogels [12] as well as hanging drop culture [13, 14].

COMMENT-3:Line 32-39: To be honest, the way the author wrote the key feature of the platform is a little tricky for me to catch on what it is the new platform can improve current limitations. I think authors can expand on the current limitations of hanging drop experiment (point 1) to link this better. I am interpreting it as a spheroid culture platform with in-built viewer that is cheap and easy to manufacture specifically to adapt to most biological studies. Consider rephrase the line 33-36 to flash this out?

REPLY: Thanks for pointing out that; the paragraph has been revised accordingly and now reads as follows (line 43-51):

The manipulation of a delicate, self-organized cellular object is challenging, laborious and often hard to accomplish. We here describe a platform for controlled cell aggregation within hanging drops. This can be easily combined with a delta-kinematic viewing stage for online monitoring of morphological aggregate characterization. In addition to that, we also included protocols for visualizing the expression patterns of markers, mounting of specimen and spatial analysis in 3D. We cared for contriving technology, which can be inexpensively and rapidly manufactured, and which in its design is easily adaptable according to individual needs. Based on this novel inventory, a rapid expedite of innovative 3D cell biological models should be accomplishable.

COMMENT-4: Methods section – image acquisition. Were all work imaged on the Leica microscope? If yes, why there is a USB camera the rig in the d-M platform? Any specific imaging parameters set for the assembled imaging platform?

REPLY: Thank you for pointing out this important issue: Images were indeed also acquired with the delta-kinematic video microscopic stage, e.g. all images shown in Figure 3. Here the delta-kinematic stage was equipped with a commercially available video scope and LED illumination positioned at low angle similar to macroscopy imaging (as shown in Fig 2). We have revised text in 2.4 and the legend of Figure 3.

(Line 112-115)

2.4 Image acquisition and Deconvolution Microscopy

Digital micrographs were acquired by means of the delta-kinematic stage equipped with a video-microscope (Bysameyee, https://www.bysameyee.
com), and with a wide-field microscope (Leica DMi8 controlled by LasX software).

(Line 252-258)

Figure 3. Hanging drop cell aggregation: Shown are time-series visualizing cell aggregation within a hanging drop, which contained 5000 cells in 20 µl medium viewed from underneath. The suspension was dispensed on the Teflon chip and incubated for indicted times within the humidified chamber. Images were acquired with the delta-kinematic stage equipped with a commercially available low-cost video-microscope (see Fig 2). Cell aggregation shown here was performed with human fetal osteoblasts, hFOB1.19 (A-D), and human osteosarcoma cells, Saos-2 (E-H). Field of view is 3 mm.

COMMENT-5: Results: SI figure 4 and the material build list should probably go into the main text in a new section. It is hard to tell what the platform is and how the parts come together. (Figure 1 does not tell much, and Figure 4 is confusing to follow)

REPLY: Following the advice of reviewer 3, Figure 4 has been split and information has been merged in Supplementary Materials, which may thus ease reproduction of the devices and comparative evaluation of the building procedure; we hope for your understanding.

COMMENT-6: Line 143-146. I am not sure how does Figure 4 protects the loss of specimens during transfer? Consider expand more.

REPLY: More information on that has been added and now reads as follows (line 165-172):

Handling and treatment of delicate cell aggregates are difficult, often resulting in destruction and loss of the specimen, in particular when attempting to transfer a spheroid by means of pipetting. We therefore designed and manufactured a device, which also warrants superimposition of a hanging drop above a microwell (Figure 4 A-I). When placing a well, which is suitable for further incubation, staining or microscopic analysis, underneath the hanging drop, and when adding further medium to a drop through the upper opening, the drop falls off. This transfers the spheroids into the microwell in a contactless fashion and thus greatly reduces the risk of damage and loss.

COMMENT-7:Figure 2, label the parts for better clarity

REPLY:

COMMENT-8: Figure 3, scale bar missing. Better to rotate the time axis 180 degrees for better reading. I am sure you can still measure the diameter of the spheroids in 48 hours for osteaoblast. Consider measuring them compare the aggregates from the osteosarcoma.

REPLY: The figure has been rotated according to the reviewer’s advice. We have also included information regarding scaling as the field of view is 3-mm, set by the lower outlet of the Teflon chip, Due to further comments on the description of Figure 3, the corresponding figure legend now reads as follows (line 252-258):

Figure 3. Hanging drop cell aggregation: Shown are time-series visualizing cell aggregation within a hanging drop, which contained 5000 cells in 20 µl medium, viewed from underneath. The suspension was dispensed on the Teflon chip and incubated for indicted times within the hu-midified chamber. Images were acquired with the delta-kinematic stage equipped with a commercially available low-cost video-microscope (see Fig 2). Cell aggregation shown here was per-formed with human fetal osteoblasts, hFOB1.19 (A-D), and human osteosarcoma cells, Saos-2 (E-H). Field of view is 3 mm.

COMMENT-9: Figure 4 – label for better clarity, missing and unclear scale bar.

REPLY: The figure has been revised also according to the advice of reviewer 3. Hence, panel A and B as well as G-I were moved to supplementary materials and split into two supplementary figure S.Figure 1.1 and S.Figure 2.1. We hope for your understanding.

COMMENT-10: Figure 5 – which cell lines are these from? Scalebar blurry.

REPLY: the cells are human origin; this has been now highlighted in the main text, as well as in the respective figure legends; scale bars have been enhanced.

COMMENT-11: Figure 5 H and J. These are the signal intensity between “stained” and ‘not stained” groups. From the aims in the introduction, authors should consider comparing signal between deconvolution vs wholemount vs histology group instead. Figure 5 H and J, words and lines are blurry, axis label missing, use better resolution.

REPLY: The authors consider deconvolution microscopy being a very good standard in the field and its validity, as it is not only widely accepted, also appears proven. We therefore refrained from providing results that comparatively show what are the pros and cons of deconvolution also in respect to other microscopic and image analysis procedures as the emphasis of this paper was on affordable and easy to accomplish experimental analyses. What we believe was necessary to demonstrate, again as this is a valid standard in our field, was to show results regarding controls. We therefore put on display an specimen, which we would not consider ‘unstained’, but a specimen that had been first treated with an antibody which is structurally related to the anti-Ki67 antibody (isotype control), and in subsequence with the same antibody bearing the same label as in the case of anti-Ki67. Figure 5 legend has been revised as follows (line 270-283):

Figure 5. A-D: Histological analysis of cell aggregates from hFOB cell line on 3-µm paraffin sections, A: hematoxylin-eosin staining; B: Masson-trichrome staining; C: immune histological analysis of the proliferation marker Ki-67 D: staining of cell nuclei with the DNA-binding fluorophore, DAPI; E,F: fluorescent imaging of a hFOB 3D whole-mount specimen at wide-field optics before deconvolution allowing direct comparison with the images derived from 3-µm histological sections shown in C and D, arrows in E point out Ki-67 positive nuclei present in the core region of spheroids; G, I: deconvoluted digital images of hFOB cell aggregates stained as 3D whole-mounts with anti-Ki-67 antibody (G) and, as a background control, with the secondary antibody only (I). H,J: Histograms depict pixel intensities along the faint yellow line drawn in the corresponding images (G,I). A threshold level was determined in unstained specimen (I). This set measure applied to discriminate signals from noise (strong yellow line in H and J) in stained specimen (G). Shown here is spatial distribution and the corresponding analysis for the proliferation marker Ki-67 (G,H) Abscizzae in G and I is the spatial distance in µm, ordinate axis shows arbitrary fluorescence units.

COMMENT-12: Line 319-322 – I am not exactly sure what f this paragraph isabout.

REPLY: Sorry for this mistake; the paragraph was generated by sloppy formatting of a preliminary version apparently still containing unwanted in-text comments. This has been now deleted.

COMMENT-13: 4 – label the parts

REPLY: We guessed that due to the position of the comment, supplementary figures were meant. Thanks for pointing out this inconsistency; figures have been merged and indications regarding building parts have been incorporated into the figure (line 501).

COMMENT-14: For the aim of the paper, what is the current go-to method for viewing hanging drops? Why not just pipette/collect and view in microscopes?

REPLY: Thank you for pointing out this deficit; we have added the following information regrading this issue (line 314-318):

Applying THHD together with online monitoring now enables studying and assessing live-kinetics and dynamics of characteristic changes of different cells types during aggregation and thereafter. Also, co-aggregation experiments, for instance with endothelial cells in order to monitor and survey vessel formation under changing conditions, become feasible. Notably, also during cultivation the addition of bioactive factors to the hanging drop through the upper THDD opening is possible in an undisturbed fashion. Ultimately, due to the simple design and manufacturing of THDD, crowded side-by-side cultivation of drop cultures and displacement of cell aggregates through automated handling is feasible.

COMMENT-15: For the set out aims of the paper, might be good to compare the cost of manufacturing to the commercially available platforms too.

REPLY: Thank you for pointing at that important aspect. We have appended a concluding paragraph to the discussions section, now reading as follows (line 365-371):

Lastly, we also focussed on establishing gear, devices and beacons that can be easily, and most importantly inexpensively produced. The most elaborate and complex machinery was the δ-M. Mechanical building blocks such as 3D-printed parts, threads, slide rails, couplings, screws magnets and couplings were around €100, electronics and motors around €200 and the video microscope was less than €100. Materials and components for the other devices were below that and thus even more cost saving. This together may help reducing the economical hurdles to step into 3Dcell biological basic research.

COMMENT-16: I am not sure why is the immune-histological analysis is placed in supplementary 5.

REPLY: We believe and actually are aware that immune staining on histological sections is often performed. We only wanted to provide a complete technical documentation also including these procedures, and therefore incorporated the protocol as a supplement. This also allows side-by-side comparisons and evaluation of histochemical and whole-mount staining.

Reviewer 2 Report

The manuscript titled “Low cost devices for three-dimensional cell aggregation, real-time monitoring microscopy, microfluidic immunostaining and deconvolution analysis” by Struber et al. reports customized devices for hanging drop cell aggregate formation, aggregate staining and imaging. The manuscript is well-constructed. Please see the following comments.

  1. The title indicates “low cost” devices, but there is no discussion of the cost of the devices in the manuscript or comparison of the cost to other cell aggregation formation and/or imaging systems. Please address this issue.
  2. Line 22. The sentence “Cell aggregates are regarded different from cells” is confusing.
  3. Line 123. “(Figure 1 D-F).” there is no image F in figure 1.
  4. Line 319-322 appear to be comments from a previous reviewer.
  5. For image acquisition and process, is the Teflon-hanging-drop-device (THDD) compatible with a commercially available regular wide-field microscope or confocal microscope? Please indicate the reason for using a custom-made microscope.
  6. Please carefully revisit the entire manuscript for grammar or typo issues. For example: in Line 302 “adher”.

Author Response

Reviewer 2

The manuscript titled “Low cost devices for three-dimensional cell aggregation, real-time monitoring microscopy, microfluidic immunostaining and deconvolution analysis” by Struber et al. reports customized devices for hanging drop cell aggregate formation, aggregate staining and imaging. The manuscript is well-constructed. Please see the following

COMMENT-1: The title indicates “low cost” devices, but there is no discussion of the cost of the devices in the manuscript or comparison of the cost to other cell aggregation formation and/or imaging systems. Please address this issue.

REPLY: Thank you for pointing at that important aspect. We have appended two paragraphs and a concluding remark in the discussion section highlighting this aspect, which read as follows (line 297-305):

We therefore took utmost care to use materials suitable for this purpose, in particular Teflon [23]. Together with methods, which are open for rapid prototyping it will support and thereby promote the production and implementation of low-cost micro-fluidic methods in most academic settings. The devices described here can be easily manufactured with commonly available tools and machines (e.g. in maker spaces at academic institutions or openly accessible engineering labs of private enterprises or communities. Also, the materials are inexpensive and can be purchased without re-strictions. Similar attempts have been communicated for specific areas of research and development in need of specifying standards for implementation of a broader use of this technology [24]

(line 322-324)                                               …

Due to the inert nature of Teflon, THDDs are re-useable many times, again reducing cost and relieving budget.

(line 365-371)                                               …

Lastly, we also focussed on establishing gear, devices and beacons that can be easily, and most importantly inexpensively produced. The most elaborate and complex machinery was the δ-M. Mechanical building blocks such as 3D-printed parts, threads, slide rails, couplings, screws magnets and couplings were around €100, electronics and motors around €200 and the video microscope was less than €100. Materials and components for the other devices were below that and thus even more cost saving. This together may help reducing the economical hurdles to step into 3Dcell biological basic research.

COMMENT-2:Line 22. The sentence “Cell aggregates are regarded different from cells” is confusing.

REPLY: Thank you for pointing out, the sentence has been revised and now reads as follows (Line 30-31):

Cell aggregates are regarded different from cells cultivated on synthetic substrates as they build up micro-tissues with interstitium that guides and influences gradients of nutrients and cellular waste [7].

COMMENT-3:Line 123. “(Figure 1 D-F).” there is no image F in figure 1.

REPLY: Sorry for this mistake. The text has been revised accordingly

COMMENT-4:Line 319-322 appear to be comments from a previous reviewer.

REPLY: Sorry for this mistake; the paragraph was generated by sloppy formatting of a preliminary version apparently still containing unwanted in-text comments. This has been now deleted.

COMMENT-5:For image acquisition and process, is the Teflon-hanging-drop-device (THDD) compatible with a commercially available regular wide-field microscope or confocal microscope? Please indicate the reason for using a custom-made microscope.

REPLY: The chips, as being described herein are indeed usable without limitations and restrictions also in other analytical devices. The main reasons for establishing the chip technology together with the microscope stage was an immanent lack of a seemingly inexpensive as easily applicable monitoring device amenable for putting the device into a completely humidified cell culture incubator having a small foot print and being light weight. We also wanted to implement means for remotely controlling monitoring and in due course experimental controls. These aspects have been described in the results section and read as follows (line 152-163):

δ-M was further equipped with a commercially available, inexpensive USB video microscope, which altogether was computer-controlled through a web-based interface. The compact size, i.e. small foot print and low weight of the video robot allowed placement and computer-aided operation within a sterile environment of a fully conditioned cell culture incubator (Figure 2 A-B), which is unattainable for a common research microscope. Alternatively, building fully conditioned incubation chambers onto a microscope stage are expensive. When placed into an incubator, the remote-controlled δ-M facilitated online monitoring of HIC/THDD-assisted cell aggregation. After forming a 20 µl hanging drop, the suspended cells sedimented in due course of 30 min (Figure 3 A). hFOB cells aggregated within 12 hours (Figure 3 B), and matured to first grow into delicate structures (Figure 3 C), before compacting into rigid spheroid after 45 h (Figure 3 D).

COMMENT-6:Please carefully revisit the entire manuscript for grammar or typo issues. For example: in Line 302 “adher”.

REPLY: Typos have been revised; thank you for pointing out and the entire manuscript has again been carefully revisited by the authors.

Reviewer 3 Report

The authors used the hanging drop method for 3D culture. 
1. The manuscript does not have an abstract!! it is so amazing 

2. The quality of figure 3 is so poor. 

3. I suggest moving figures 4A,B,G,H to supplementary. 

Author Response

Reviewer 3

The authors used the hanging drop method for 3D culture.

COMMENT-1: The manuscript does not have an abstract!! it is so amazing

REPLY: During submission, which also included to copy the abstract into a text box, we were later asked to bring the manuscript in to a given format. The template did not ask for an abstract section and did not include an abstract; that is why we omitted the abstract. In the revised version, we now included the abstract as a first section, hopefully now not offending the publisher’s requirements.

COMMENT-2: The quality of figure 3 is so poor.

REPLY: He images have been acquired from hanging drops, which had been illuminated with an LED positioned at a low angle above the Teflon LED. Hence, the light shines through the upper opening of the hanging drop device. Thus, the cells, which sediment down to the apex of the hanging drop (i.e. a spherical surface) are illuminated by reflecting light bundled within the drop. It was impossible to remove background gradient and light reflection (also polarized light did not solve this issue). Let us point out, that due to this physics, imaging of a specimen sitting within a hanging drop is difficult, and what we show in Figure 3 is the best what we actually could achieve. We hope we could explain why we think it is not poor quality but maybe not up to the expectations of an observer, who is used to watch images taken from thin specimen with the aid of a of microscopic under Abbe optics and Köhler illumination. This here is comparable to imaging of embryos by means of a dissection scope, where to the best of our experience, the same issues of reflecting light and gradient background apply.

COMMENT-3: I suggest moving figures 4A,B,G,H to supplementary.

REPLY: The figure has been revised also according to this advice of reviewer 3. Panel A and B as well as G-I were moved to supplementary materials and split into two supplementary figure S.Figure 1.1 and S.Figure 2.1.

Round 2

Reviewer 1 Report

Authors addressed most of the comments.

Figure 4 C and D - scale bar too blur to read.

Figure 5 picture quality should be improve, the words are not readable (especially H and J). It is almost impossible to read the axes and the lines.

Author Response

Thank you for pointing out the short-comings. 

We have revised  Figures 4 and 5 according to your advice:
- Figure 4: scale bars have been enhanced and the information regarding lenght (in all cases 100 µm) can be now found in the figure legend
- Figure 5:  labels for the axes of the graphs shown H and J have been enlarged to a size still fitting within the panel outlines; we hope that this can now be fully appreciated by the readers   

Reviewer 2 Report

Comments have been addressed in the revised version. 

Author Response

The manuscript has been again read and carefully revised regarding wording, spelling and gramar:

Cody Wickstrom and Eric Lutsch are native speakers (US English)
Sabrina Marozin and Martin Fischlechner were educated in the UK and worked there for many years at academic institutions
Günter Lepperdinger has worked many years in the US and  serves at Elsevier and Karger since 2004 as an Editor for Scientific Journals . 

The authoring team is confident that the English text is at a high standard and apt for publication in an international scientific journal 

Reviewer 3 Report

The abstract has been added to the manuscript. the text has been revised. 

Author Response

THank you for your help in improving the publication